# Role of the Cyclooxygenase Pathway in the Association of Obstructive Sleep Apnea and Cancer

**DOI:** 10.3390/jcm9103237

**Published:** 2020-10-10

**Authors:** César Picado, Jordi Roca-Ferrer

**Affiliations:** 1Hospital Clinic, Department of Medicine, Universitat de Barcelona, 08036 Barcelona, Spain; 2Institut d’Investigacions Biomèdiques August Pi i Sunyer (IDIBAPS), 08036 Barcelona, Spain; jrocaf@clinic.cat; 3Centro de Investigación Biomédica en Red de Enfermedades Respiratorias (CIBERES), Instituto Carlos III, 28029 Madrid, Spain

**Keywords:** cancer, cyclooxygenase, obstructive apneas, prostaglandin E2, prostaglandin receptors, sleep

## Abstract

The objective of this review is to examine the findings that link obstructive sleep apnea (OSA) with cancer and the role played by the cyclooxygenase (COX) pathway in this association. Epidemiological studies in humans suggest a link between OSA and increased cancer incidence and mortality. Studies carried out in animal models have shown that intermittent hypoxia (IH) induces changes in several signaling pathways involved in the regulation of host immunological surveillance that results in tumor establishment and invasion. IH induces the expression of cyclooxygenase 2 (COX-2) that results in an increased synthesis of prostaglandin E2 (PGE_2_). PGE_2_ modulates the function of multiple cells involved in immune responses including T lymphocytes, NK cells, dendritic cells, macrophages, and myeloid-derived suppressor cells. In a mouse model blockage of COX-2/PGE_2_ abrogated the pro-oncogenic effects of IH. Despite the fact that aspirin inhibits PGE_2_ production and prevents the development of cancer, none of the epidemiological studies that investigated the association of OSA and cancer included aspirin use in the analysis. Studies are needed to investigate the regulation of the COX-2/PGE_2_ pathway and PGE_2_ production in patients with OSA, to better define the role of this axis in the physiopathology of OSA and the potential role of aspirin in preventing the development of cancer.

## 1. Introduction

Obstructive sleep apnea (OSA) is characterized by recurrent episodes of upper airway[M1] obstruction occurring during sleep, leading to chronic intermittent hypoxia and sleep fragmentation. Evidence accumulated during the last three decades has demonstrated the association of OSA with hypertension, metabolic dysfunction, cognitive dysfunction, and coronary and cerebrovascular artery disease that results in excess mortality [1]. In addition, recent observations support that the excess in mortality detected in OSA may also be the consequence of an increased incidence of malignant tumors in these patients [2,3,4,5,6].

The objective of this review is to analyze the mechanisms that seem to be involved in the predisposition of patients with OSA to develop malignant tumors, as well as the potential role that activation of the cyclooxygenase (COX) pathway can play in this association.

## 2. Obstructive Sleep Apnea and Cancer

### Obstructive Sleep Apnea and Cancer: Epidemiological Studies

The association between OSA and cancer has been reported by studies carried out in various cohorts using sleep laboratory-based populations studies.

The Wisconsin cohort showed that subjects with severe hypoxemia during sleep were 8.6 times more likely to die of cancer than those without OSA. When an hypoxemia index was used as a continued variable, an increase in one log-unit in the index was associated with an adjusted hazard ratio of cancer mortality of 1.9 (95% CI, 1.3–2.9; *p* = 0.002) [2]. The Spanish authors found that cancer incidence and mortality risk augmented with increasing levels of OSA and the hypoxemic index [3]. When the subgroup of 527 patients with a diagnosis of cancer was statistically analyzed, the hypoxemic index was significantly associated with cancer mortality after adjusting for confounders [4]. In an Australian study, moderate-severe OSA was significantly associated with cancer mortality and incident cancer [5]. The relationship between sleep apneas with Central Nervous System (CNS) cancer risk was examined using an administrative database involving 23,055 patients with OSA and a comparison age-and gender-matched non-OSA group (*n* = 69,165) [6]. The risk of CNS malignant tumors was 1.5 times higher in OSA patients than in the control group over a two-year period of follow-up [6]. Using the same database, the risk of developing breast cancer in the OSA sample was two-fold compared with the age-matched control women [7]. A recent study provided evidence supporting the carcinogenesis role of OSA in breast cancer [8]. Interestingly, the study shows that the genetic determination risk of OSA has a casual effect on the elevated risk of breast cancer in OSA patients [8]. In a recent study the simultaneous occurrence of lung cancer and OSA was found in 43 patients among about 500,000 patients admitted to a hospital in one year [9]. No significant difference in lung cancer incidence was found among OSA patients and hospital population. However, survival of patients with lung cancer was related with the severity of OSA. Patients with moderate to severe OSA had a lower overall survival than patients with mild OSA [9]. Measures of melanoma aggressiveness were found to be associated with the severity of nocturnal oxygen desaturation indices [10]. Taken together these studies suggest that OSA may have an impact on the accelerated progression of some malignant tumors.

The reported association between OSA and cancer found in some studies has been questioned by other studies that have not been able to demonstrate this relationship [11,12,13,14]. Using the Copenhagen City Heart Study cohort, 5894 subjects were selected to assess the relationship between symptoms of sleep disordered breathing (SDB) and the risk of cancer [11]. Snoring, breathing cessation during sleep, and daytime sleepiness were used to characterize SDB. There were no associations between snoring, breathing cessations, or total number of SDB symptoms and total cancer incidence. However, high levels of daytime sleepiness were associated with a higher cancer risk among subjects younger than 50 [11]. The negative conclusion of the study has been questioned due to the limited sensitivity and specificity of symptom-self report to characterize SDB which can result in misclassification [15]. In another study, cancer diagnosis was identified using a cancer registry [12]. Two outcomes were examined: prevalent cancer at the time of the diagnostic sleep study (baseline), and diagnosis of cancer (cancer incidence) among patients free of cancer at baseline. Severity of OSA was not independently associated with either prevalent or incident cancer after the adjustment for demographic and comorbidities [12]. Another study assessed the association of OSA with different types of cancer [13]. All OSA diagnoses included in an administrative health insurance database involving 77 million subjects were identified and 1:1 matched demographically, state of residence and comorbidities with non-OSA subjects. To examine the potential ability of OSA to alter the clinical course of cancer, another cohort of patients (*n* = 700,000) with a primary diagnosis of cancer was identified, and the risk of metastatic disease or cancer mortality was determined as a function of the presence (*n* = 30,000) or absence of OSA adjusting for concurrent comorbidities. The incidence of any type of cancer was found to be similar in OSA and matched controls. After adjustment for comorbidities only the incidence of melanoma, and kidney and pancreatic cancer, remained significantly elevated. In contrast, the incidence of breast, colon, rectum, and prostate cancers was significantly lower in OSA patients when compared to control cohorts. The presence of OSA was associated with similar or lower risk of developing metastasis than cancer patients without OSA. Similarly, mortality rates for any cancer type were either similar or lower in the OSA cohort than in non-OSA matched group. The study concluded that OSA appears to increase the risk for only a selective number of cancer types, and does not appear to be associated with an increased risk of metastatic cancer or cancer-related deaths [13]. A recent study found that the incidence of cancer was higher among patients with OSA than in the control group. Similar to the previous study, the high incidence was observed in some cancers (kidney, melanoma, breast), while risk for other cancers (lung, colorectal) was lower with respect to the control group [14].

Taken together the contradictory results reported by the various population-based studies on the association between OSA and cancer reveal the complexity of the problem and highlight the need for further population-based cohort studies with OSA screening, cancer diagnosis, and prolonged follow-up.

## 3. Cancer and Immunity

Malignant transformation may occur frequently in the body, but according to the immune surveillance theory, immune cells continuously recognize and destroy nascent tumor cells. However, new variants of neoplastic tumor cells continuously arise with the potential to evade the immune surveillance that finally result in tumor establishment and progression, what is called the immunoediting process [16,17,18]. The immunoediting process encompasses three phases: elimination, equilibrium, and escape [16]. In the first phase immune surveillance efficiently eliminates abnormal cells. Progression to the next phase of immunoediting occurs if an abnormal cell evades immune destruction. The progeny of such a cell might allow progression to the second phase called “immune equilibrium”. During this phase, unlike the next, the tumor does not change in size significantly, and remains relatively well controlled. With the evolution of further mutation and metabolic changes, the cancer cells can finally evade the host’s immune reactions. In this third phase, called “immune escape”, the cancer cell progeny will efficiently evade the immune system’s imposed equilibrium, and the tumor will begin to grow, invade, and metastasize [17,18].

Tumor cell immune evasion is a complex process that involves, among others, alteration of normal T helper (Th) Th1 and Th2 lymphocytes, and regulatory T-cells (Tregs) immune responses, impaired cytotoxic activity of CD8+T cells and natural killer (NK) cells, a defective dendritic cell (DC) functioning, a shift of macrophages (M) from M1 (Th1 response) to M2 (Th2 response) polarization states, and enhancement of immunosuppressive regulatory myeloid-derived suppressor cells (MDSCs) [16,17,18].

### 3.1. Immunoediting: Role of Th1, Th2, Tregs, and Th17

The progression of human cancer is associated with a shift in Th1 to Th2 immune responses and enhancement of Tregs responses [1]. Th lymphocytes play a key role in the adaptive immune system exerting a wide spectrum of biological functions. CD4+T cells regulate both the cytotoxic cellular immune response and B cell-dependent antibody production. CD4+T cells can differentiate into various subpopulations of T cells with specific functions and properties [19]. CD4+T helper cells deriving from the thymus differentiate at the periphery in response to antigen stimulation [19]. The first classification divided CD4+ effector cells into two subsets, Th1 and Th2 [20]. Th1 cells are induced in response to pathogens, such as viral infections, and are characterized by the production and release of interferon gamma (IFN-γ). Th1 cells also promote the activation of macrophages that are efficient against intracellular pathogens. Th2 cells are basically involved in humoral immune response and provide help to B cells to produce class-switched antibodies [21]. Th1 lymphocytes exert an antitumor effect by releasing tumor necrosis factor α (TNF-α), and IFN-γ, while Th2 cells mainly produce interleukin (IL) 4 (IL-4) and thereby promote tumor growth by inhibiting the host’s immune system [1].

Studies carried out in the last decade, reported that more functional subsets of T helper cells can be induced by various stimuli in vivo and in vitro. CD4+T cells exposed to transforming growth factor beta (TGF-β) and certain interleukins such as IL-6 differentiate into CD+T cells producing IL-17, and were recognized as a distinct subset of Th cells [21]. Another subset of CD4+T lymphocytes are the (Tregs). Tregs have a role in regulating other cells in the immune system, thereby maintaining self-tolerance and homeostasis [22].

The developing tumor induces an immune reaction driven by IFN-γ producing CD4+T cells that are recruited to the tumor and induce cell death [23]. Th1 subsets play an important role in the antitumor response, producing inflammatory cytokines and assisting the cell-mediated killing of tumor cells [23]. These responses, however, can be suppressed by Treg cells that are also recruited by the growing tumor [24]. Tregs are potent inhibitors of antitumor immunity and at the early stages of the process are concentrated in the human tumor mass, locally inhibiting the effector’s immune responses and allowing the tumor to progress [25]. At later stages, Tregs can be up regulated systemically, suppressing the immune protection against metastases in human tumors [26].

Studies on the role of Th17 cells in cancer have reported controversial results. It has been shown that Th17 cells numbers infiltrating human tumors are higher in comparison with surrounding tissues implying a specific role in tumor development [27]. Th17 cells take part in local inflammation producing IL-17 and IFN-γ and can, therefore, promote the inflammation-dependent tumor cell growth [27]. Such an accumulation of Th17 cells was associated with improved patient survival in some human cancer types and with poor prognosis in other types [28]. Moreover, Th17 cells and IL-17A cytokines were shown to promote angiogenesis in human malignant cells [29]. However, other cytokines also produced by Th17 cells (IL-17F, IL-21, and IL-22) shared anti-angiogenic properties [30,31,32].

### 3.2. Immunoediting: Role of CD8+T and NK Cells

Most cancer cells express antigens that can be recognized by T cells of the host immune system that results in the expansion of CD8+T cells specific for tumor-associated antigens [33]. CD8+T lymphocytes kill malignant cells upon recognition by the T-cell receptor (TCR) of specific antigenic peptides expressed on the surface of target cells by human leukocyte antigen class I (HLA-I)/beta-2-microglobulin (ß2 m) complexes with the contribution of numerous costimulatory molecules and receptors. In addition, these lymphocytes can also direct their cytotoxic activity against malignant cells by recognizing tumor-specific antigens encoded by mutated genes (neoantigens) [33,34,35]. However, cancer progression is seen in most of these patients, an observation that supports that the tumor cells can escape immune-mediated destruction [33]. The immune failure implies that the immune inhibitory mechanisms within the tumor microenvironment prevails over the T cells cytotoxic effects. Indeed, the microenvironment of these tumors is characterized by the presence of various molecules such as PD-L1 (programmed death ligand 1), indoleleamine-2,3-dioxygenases (IDO), and cells such as Treg that are associated with immune tolerance [36].

NK cells can be found in the tumor microenvironment and their presence has been associated with favorable prognosis in cancer patients [37]. NK can be activated in response to DNA damage of tumor cells, which leads to increased expression of ligands on tumor cells for NK cell-activating receptor. Some cytokines such as IL-12, IL-15, and IL-18 activate NK cells for tumor control [38]. Activated NK cells kill tumor cells directly by releasing perforin and granzymes from cytoplasmatic granules and secreting IFN-γ [39]. NK cells can indirectly contribute to tumor destruction through the activation of CD8+ cytotoxic T cells and also by promoting the differentiation of Th1 CD4+T cells through production of IFN-γ [40].

### 3.3. Immunoediting: Role of Dendritic Cells

DCs possess the capacity for switching between tolerogenic and effector/cytotoxic phenotypes [41]. Immature DCs are cells prone to develop tolerogenic responses while activated dendritic cells are more efficient at promoting effective responses by T cells. DCs, including myeloid dendritic cells (mDCs) and plasmacytoid dendritic cells (pDCs) participate in tumor immunosurveillance by using various mechanisms [41]. pDCs are the principal producers of type I interferons (IFNs) in response to microbial and viral infection [42]. Type I IFN, in addition to its antiviral functions, is critical for preventing the establishment of tumors through its ability to enhance both innate and adaptive immune responses [43]. Furthermore, mDCs secrete large amounts of IL-12 upon activation and are therefore important for generating immune responses against pathogenic organisms or to suppress neoplastic cell growth [44]. Dendritic cells can also process and present lipid antigens. Lipid antigens complexed with CD1d molecules on the surface of DCs activate natural killer T cells [45]. Mature DCs can induce tumor-specific cytotoxic activity that inhibits tumor growth [46]. Malignant cells release factors in the tumor microenvironment which suppress maturation and antitumor activity of dendritic cells [41,47,48]. The concentration of cytokines that favors dendritic cell development and function, like the granulocyte-macrophage colony stimulating factor (GM-CSF), IL-4, IL-12, and IFN-γ, is very low in the tumor environment, while factors that suppress dendritic cells, such as IL-6, IL-10, vascular endothelial growth factor (VEGF), and transforming growth factor β (TGF-β), are found in higher levels [41,48]. As a result of this altered cytokine balance, the number of activated dendritic cells is very low in tumor areas and all steps of dendritic cell development, migration, and activity may present significant defects [41,49,50,51].

### 3.4. Immunoediting: Role of Tumor-Associated Macrophages

When blood monocytes are recruited into tissues they can differentiate into distinct, mature macrophages and exert specific immunological functions. Monocytes are attracted and recruited to the tumor by chemokines such as colony stimulating factor 1 (CSF-1) which are secreted by tumor cells. Once monocytes have reached the tumor tissue they can proliferate and differentiate into distinct, mature macrophages and exert a specific immunological function [52,53]. Macrophages may represent up to 50% of the tumor mass. These tumor-associated macrophages (TAM) play a major role in the links between inflammation and cancer [52,53]. Macrophages can be activated to either M1 (Th1 response) or to M2 (Th 2 response) polarization depending on the microenvironment stimuli [52,53,54]. IFN-γ and TNF-α have been reported to induce the activation of M1 phenotype, while TGFβ, IL-6 and IL-10, IL-4 and IL-13 play a major role in the induction of M2 macrophages [52,53,54]. M1 macrophages produce IL-12^high^ and IL-10^low^ cytokine profile and contribute to tumor control [51,52]. In contrast, the M2 phenotype has an IL-12^low^, IL-10^high^ cytokine profile that exerts a pro-tumorigenic role [52]. M2-like TAM produces PGE_2_ and TGF-β that can suppress anti-tumor T cell responses [55]. Moreover, M2-like TAMs also recruit Tregs that results in an enhanced immune suppressive effect [56]. All in all, the pro-tumorigenic activities of TAM cover various key features of neoplastic cell proliferation, migration and metastasis, and survival in hypoxic conditions due to suppression of anti-tumor immunity [30].

### 3.5. Immunoediting: Role of Myeloid-Derived Suppressor Cells (MDSC)

Myeloid cells represent a highly heterogeneous population. Mononuclear myeloid cells include monocytes which under inflammatory conditions differentiate in tissues to macrophages and dendritic cells [57]. Granulocytic myeloid cells include populations of neutrophils, eosinophils, basophils, and mast cells [57]. Expansion and activation of neutrophils and monocytes (myelopoiesis) in response to pathogenic stimuli such as pathogen-associated molecular patterns (PAMP) or danger-associated molecular pattern (DAMP) molecules, is a crucial mechanism to protect the host [57]. This activation results in the release of inflammatory cytokines, production of reactive oxygen species (ROS), and phagocytosis. The defensive response is short-lived and terminates when the injury abates. However, if the aggression persists due to a chronic inflammation or infection a modest but persistent myelopoiesis can be maintained. Myeloid cells generated under these conditions acquire a distinct genomic and biochemical profile and functional activity and they develop the ability to suppress various types of immune responses [57]. Various studies reported the accumulation of these immuno-suppressive cells in tumors. Different phenotypic criteria were used to characterize the cells. Based on the myeloid origin and their immuno-suppressive function, MDSCs were proposed to identify these cells [57]. However, it became evident that the cells were heterogeneous [57].

In mice, two subsets of MDSCs, polymorphonuclear (PMN-MDSCs) and monocytic (M-DSCs) have been characterized. In most cancer types PMN-MDSCs represent more than 80% of all MDSCs. In humans MDSCs are mostly found in blood, tumors, and bone marrow [57,58]. A two-signal model has been proposed to account for MDSC recruitment, expansion, and activation in tumors [59]. The first group of signals is mostly driven by tumor-derived growth factors such as STAT3, IRF8N, Notch, and NLRP3 among others [57,59]. The second group of signals is mainly composed of factors produced by the tumor stroma and includes the NF-κB pathway, STAT1 and STAT6 [57,59]. MDSCs contribute to cancer immune evasion via a complex and varied mechanisms including among others: (1) a deprivation of amino acids arginine and cysteine, which are essential for T cell proliferation and anti-tumor reactivity [59,60]; (2) an enhanced production of IL-10 and TGF-β1 that results in the inhibition of various immune effector functions [60,61,62]; (3) an increased production of nitric oxide and ROS that promote apoptosis of NK and T cells [59,63,64]; (4) a down-regulated anti-tumor T cell-mediated reactivity resulting from the up-regulation of the expression of the programmed death-ligand 1 (PD-L1) [59,60,64,65], and increased release of angiogenic mediators that stimulate tumor neovascularization [59,66].

A positive correlation of MDSC numbers in peripheral blood with human cancer stage and tumor burden has been found in colorectal carcinomas, breast, bladder, thyroid, and melanoma [57] Moreover, elevated numbers of MDSC in the blood are an independent indicator of poor outcomes in patients with solid tumor [57].

### 3.6. Immunoediting: Role of Angiogenesis and Hypoxia

Angiogenesis refers to the physiological process in which new vessel cells grow from pre-existing vessels [67]. Angiogenesis is vital for processes such as wound healing, but it is also fundamental for solid tumor growth and metastasis [68]. Tumor cells must be within a certain distance of a perfused blood to receive sufficient oxygen and nutrients to survive and proliferate. Tumor sizes are limited to 2 mm in diameter without neovascularization [68]. VEGF is the predominant mediator of tumor angiogenesis. Tissue hypoxia due to inadequate blood supply has been reported to develop very early during tumor development [67,68,69]. Malignant cells can also produce VEGF to promote angiogenesis even before the formation of a visible tumor. However, due to the exaggerated growth rate of tumor cells, solid tumors still experience hypoxia despite the activated vessel formation [68,69,70,71]. The key mediator involved in the transcriptional hypoxic responses is hypoxia-inducible factor (HIF). HIF is a dimeric protein composed of an O_2_-regulated α subunit (HIF-1α, or HIF-2α) and a constitutively expressed HIF-1β subunit. Hypoxia blocks the capacity of T cells to expand and perform their cytotoxic effector functions [70]. Under conditions of hypoxia immune-suppressive cells TAM, MDSC, and Tregs are recruited, expanded, and phenotypically programmed to enhance their suppressive capacity [70,71]. Moreover, factors secreted by hypoxic cell tumors suppress NK cell functions [71]. Accumulated evidence also indicates that activation of HIF-1α-dependent signaling at the primary tumor promotes metastatic dissemination and colonization [70,71]. Overexpression of HIF-1α is associated with an aggressive phenotype and increased mortality in many cancers. All in all, these observations support the notion that tumor hypoxia plays a crucial role in establishing and maintaining an immune privilege environment for tumor evasion [71].

## 4. Cyclooxygenase Pathway and Cancer

### 4.1. Cyclooxygenase Pathway

COX enzymes 1 (COX-1)-and 2 (COX-2) catalyze the rate-limiting step in the production of eicosanoids from the arachidonic acid (AA) present in the cell membrane. COX-1 is constitutively expressed in most tissues and is involved in housekeeping functions that contribute to maintain tissue homeostasis. By contrast, COX-2 is an early inducible gene normally absent from most cells but rapidly and highly inducible at sites of inflammation [18,72].

COX enzymes convert AA to the precursor molecule prostaglandin H_2_ (PGH_2_). PGH_2_ can then be converted to one of the five prostanoids including four prostaglandins (PG) such as PGE_2_, PGF_2α_, PGD_2_ and PGI_2_, and thromboxane A_2_ (TxA_2_) through specific synthase enzymes. PGs are among the major players involved in the regulation of inflammatory responses by activating and regulating numerous signaling pathways [72].

Among the numerous metabolites formed from AA, PGE_2_ is the one that has been most thoroughly investigated in cancer. PGE_2_ plays a central role in the regulation of multiple biological processes under both normal and pathological conditions [18,72]. There are several specific synthases involved in the synthesis of PGE_2_, one is cytosolic (cPGES) and another is microsomal or membrane bound (mPGES). cPGES is predominantly coupled to COX-1, and mPGES is preferentially linked to COX-2 and exists in two isoforms, mPGES-1 and mPGES-2. Similar to COX-2, the expression of mPGES-1 can be induced by proinflammatory signals. mPGES-1 is the synthase that is primarily responsible for increasing PGE_2_ release during inflammation. Once PGE_2_ is produced, it is exported into the extracellular microenvironment where it exerts its biological effects in an autocrine and paracrine manner through its four cell surface prostaglandin receptor subtypes (EP1, EP2, EP3, and EP4) [18,72,73].

EP receptors are classical, plasma membrane-expressed, G-protein-coupled receptors [73]. Recent studies indicate that EP receptors are also expressed at other subcellular locations. Each receptor has distinct tissue and cellular localization and is coupled to different intracellular signaling pathways. The EP1 receptor couples with the Gα_q_ protein subunit that activates the phospholipase C (PLC)-inositol, this phosphated (IP3) pathway, and its activation results in the release of intracellular calcium. EP2 and EP4 receptors are linked to G-stimulatory (Gα_s_) proteins and activate adenylate cyclase (AC) increasing cAMP levels which results in the activation of protein kinase A (PKA). The EP3 receptor is unique in that it exists as various alternative splice variants. As a result, EP3 is capable of stimulating or inhibiting cAMP (by stimulating or inhibiting AC), as well as stimulating calcium mobilization possibly via PLC. However, the major EP3 splice variant is the one that is coupled to an inhibitory (Gi) protein, and hence the major outcome of EP3 receptor signaling is the inhibition of AC [73].

Interactions between PGE_2_ and its receptors are thought to be dependent on specific receptor cell distribution, level of expression, and on variation in binding affinities. These differences explain the ability of PGE_2_ to mediate highly varied biological effects in many different cells and tissues types [73].

### 4.2. COX-2 Pathway: Role in Tumorigenesis

Initially recognized in the context of colorectal cancer, COX-2 has been identified in many human cancers and precancerous lesions [72]. Numerous studies have reported that COX-2 is over expressed in tumor cells including lung [74,75,76,77], colorectal [75,76,77,78], mesothelioma [77], hepatocellular carcinoma [78], pancreas [79], breast [74,80], melanoma [81], and others [72,82]. COX-2 has also been associated with cancer metastasis [18,72]. COX-2 overexpression is usually associated with the concomitant expression of mPGEs-1, the terminal synthase that leads to the preferential production of PGE_2_ [18]. Most of the modulator effects of PGE_2_ on immune regulation take place as a result of signaling through the EP2 and EP4 receptors [83].

### 4.3. Role of Prostaglandin E2 in the Immunoediting Process

PGE_2_ inhibits proliferation and effector functions of CD4+ and CD8+T cells and promotes their differentiation in Treg in human cancer cells [84,85]. PGE_2_ inhibits the production of TNFα and IL-12 which results in suppressing type-1 immune responses. Proliferation of Th1 T cells is inhibited through the EP2 receptor in mice macrophages [86].

Induction of Treg is associated with the level of expression of COX-2 in human cells [87]. It has been shown that Treg cells contribute to immunosuppression in human cancer by inhibiting Th1 cells in a COX-2-dependent manner [88]. The suppressive activity of Treg cells is driven by the expression of the forkhead/winged helix transcription factor (FOXP3) gene in mice cells [89]. COX-2 inhibitors attenuate Treg cell activity and Foxp3 expression in tumor-infiltrating lymphocytes, enhancing antitumor response that results in the delay of tumor growth [90].

PGE_2_ may exert opposite effects on Th17 cells. Some studies reported that the generation of the Th17 subset derived from CD+T cells exposed to TGF-β and IL-6, is suppressed by PGE_2_ through both EP2 and EP4 receptors in mice cells [91]. In contrast, PGE_2_ can also enhance Th17 cells differentiation in the presence of IL-1β and IL-23 also through EP2 and EP4 receptors in human and murine cells [92].

With increasing tumor burden, the NK cell activity decreases [93,94]. PGE_2_ inhibits the potential of NK cell to migrate, exert cytotoxic effects, and secrete IFN-γ [93]. When COX-2 is inhibited by indomethacin the NK cell cytotoxicity is restored in human and murine cells [93,94]. A recent study demonstrated that the major NK receptors: NKp30, NKp44, and NKp46 are inhibited by PGE_2_ in mice tumor cells [95]. Although the NK cells express all PGE_2_ receptors, the ability of PGE_2_ to inhibit NK cells is by acting on EP2 and EP4 receptors [95,96]. COX-2 inhibitors and the EP4 receptor antagonist contribute to re-establish the NK functions that are critical to the control of metastasis. In addition, murine and human mammary tumor cells treated with COX-inhibitors are more sensitive to the lytic capacity of NK [97,98].

PGE_2_ modulates DC differentiation, maturation, and function [99,100]. PGE_2_ promotes tumor growth by suppressing tumor-infiltrating DC activation in a mouse model and cancer cell lines [101]. PGE_2_ inhibits DC differentiation and maturation using various regulatory pathways such as the retinal dehydrogenases enzymes which are involved in the synthesis of retinoic acid that is required to modulate DC function in mice and humans [102]. PGE_2_ can also promote the differentiation of human DCs into monocytic MDSCs, cells that play a central role in cancer immune evasion [103]. Meanwhile, PGE_2_ enhances the production of endogenous IL-10 which down-regulates bone marrow-derived dendritic cell functions via EP2/EP4 receptor subtypes [99]. In addition, PGE_2_ inhibits the secretion of IFN-α by activated human PDCs via EP2/EP4 which results in the induction of Th2 cytokine secretion and reduction of Th1 cytokine secretion [104]. Taken together, these observations support that targeting PGE_2_/EP2-EP4 receptor signaling may be a powerful mechanism for modulating DC activity.

Although M2-like TAMs produce PGE_2_ that can suppress anti-tumor T cell responses [105], there is only limited and indirect evidence on the role of PGE_2_ in the regulation of macrophage polarization to the M2 phenotype during tumor initiation and progression. In a murine breast cancer model, COX-2 inhibition shifts the phenotype of tumor-associated macrophages from M2 to M1 and suppressed metastasis [106]. In a colon tumor implantation in a murine model, the overexpression in tumor tissue of 15-PGDH, an enzyme involved in PGE_2_ catabolism, resulted in the differentiation of intratumoral myeloid cells from M2-oriented TAMs to M1-oriented macrophages [107].

In tumor-bearing animals, PGE_2_ was shown to be a potent immune-suppressive mechanism produced by MDSCs [108]. COX-2 inhibition by non-steroidal anti-inflammatory drugs prevents the local and systemic expansion of MDSCs [109]. Similarly, recent studies have shown that MDSC-derived PGE_2_ is essential for maintaining the suppressive activity of MDSCs in human cancers [103,110]. PGE_2_ is also known to augment IL-10 induced activation of STAT-3 which has been proposed as a key regulator of the suppressive functions of MDSCs by increasing arginase-I production [111,112].

In the course of solid tumor development, the avascular tumor mass becomes dependent on angiogenesis for progression [68,69,70,71]. Over-expression of COX-2 in human cancer cells induces the production of VEGF which is basic in stimulating the formation of new blood vessels, a requirement for tumors to develop beyond a few millimeters in size [113,114,115]. Several studies have demonstrated the important role played by the COX-2/mPGES-1/PGE_2_ pathway in angiogenesis [113,114,115].

Hypoxic stress plays a key role in tumor promotion and immune evasion by controlling angiogenesis and favoring immune suppression [68,69,70,71]. PGE_2_ has been reported to stimulate VEGF expression in human cancer cells through the activation of the HIF-1α pathway [116]. The pro-angiogenic effects of the COX-2/mPGE-1 pathway can be inhibited by NSAIDs in human and animal cancer cells lines and rescued by adding exogenous PGE_2_, suggesting that COX-2 derived PGE_2_ is, at least in part, responsible for the pro-angiogenic effects of COX-2 over-expression [114,117,118] (Appendix A).

## 5. Aspirin in the Prevention of Cancer

Taking into account the numerous observations collected from in vitro and animal model studies showing the relevant regulatory role played by the COX/PGE_2_ pathway in tumorigenesis, it seems logical to expect that the use in real life of aspirin and other NSAIDs will exert both preventive and therapeutic effects on cancer. Effectively, a large body of epidemiological evidence suggests an association between regular use of aspirin and reduced risk of incidence and death from human colorectal, esophageal, gastric, biliary, and breast cancer [119,120,121,122,123].

The first studies showing the protective effect of aspirin were reported in colorectal cancer [119,120]. Observational studies, randomized controlled trials, and data metaanalysis of studies using aspirin in the prevention of cardiovascular events collectively provide compelling evidence of the chemopreventive effect of aspirin against cancer development [121,122,123].

These studies also showed that the effect in the reduction in cancer incidence becomes apparent several years after aspirin treatment has been initiated. Interestingly, the results of some of these studies indicate that the detectable benefits were seen at low daily doses (75 mg–100 mg), and that higher doses did not increase the preventive effects of aspirin. A significant reduction in the incidence of colorectal cancer was reported in association with alternate-day 100 mg of aspirin versus placebo [124]. The protective effects of low-dose aspirin appear to result from both prevention of tumor development and anti-metastatic action [121,122,123].

The efficacy of aspirin to prevent cancer appears to be related to the level of COX-2 expression and PGE_2_ production [125,126]. Long-term aspirin use was associated with a reduced risk of developing colorectal cancer when COX-2 was overexpressed, whereas no effect was seen with weak or absent COX-2 expression [125]. Similarly, high levels of urinary metabolites of PGE_2_ predict efficacy of aspirin chemoprevention [126].

The use of aspirin to prevent the development of cancer is limited by the risk of serious side effects, in particular gastrointestinal bleeding [127]. Several studies are currently being performed to clarify the risk–benefit ratio of low dose of aspirin in the prevention of cancer [128].

## 6. Mechanisms of Tumorigenesis in OSA: Role of Cyclooxygenase Pathway

Studies carried out in animal models and cancer cells lines have demonstrated that both sleep fragmentation and IH promote tumor cell growth and invasiveness by modulating some of the mechanisms involved in the immunoediting and angiogenesis [129,130,131,132,133,134,135,136,137].

The tumorigenic regulatory activities of TAMs (M1 and M2) are critically important in hypoxic conditions [131]. Two studies have reported that sleep fragmentation and IH in mice increased the number of TAMs expressing M2 macrophage markers [131,132]. A recent study investigated the effects of sleep fragmentation and IH in the regulation of CD8+T cytotoxic cells, Tregs, and cancer stem cell populations in the murine model of sleep apnea. The number of CD8+T cells was significantly reduced in IH-exposed mice, the proportion of Tregs was increased in the tumors of mice exposed to both IH and sleep fragmentation, and the numbers of cancer stem cells were also increased in intermittent and sleep fragmentation tumors [133]. Similarly, tumors in mice exposed to cyclic hypoxia showed enhanced angiogenesis and increased expression of VEGF, an effect that can also contribute to facilitate tumor growth and metastatic dissemination [134,135,136]. IH activates the Wnt/β-Catenin signaling pathway and contributes to accelerate cancer growth in a mouse model [137]. Hypoxia-induced cell proliferation and invasion via Wnt/β-Catenin has been recently reported in a mouse model of lung cancer [137].

Exosomes are defined as extracellular vesicles that are released from cells upon fusion of an intermediate endocytic compartment, the multivesicular body, with the plasma membrane [138]. Exosomes are thought to play a relevant role in intercellular communications by the transmission of lipids, proteins, mRNA, miRNA, and DNA [138]. It has been reported that exosomes released from the neoplastic cells contribute to repressing immune surveillance of tumors [138]. A recent study evaluated the effects of IH in the release of tumor-promoting exosomes in the circulation in mice injected with TC1 lung carcinoma compared to mice not bearing TC1 cells [139]. Exosomes were also isolated from patients diagnosed with OSA at baseline and after CPAP treatment for 6 weeks. Exosomes from mice and patients were co-cultured with mouse TC1 and human adenocarcinoma cells respectively. Circulating exosomes released under IH conditions from tumor-bearing and non-tumor-bearing mice significantly promoted TC1 malignant properties. Similarly, exosomes isolated from untreated patients with OSA significantly enhanced proliferation and migration of human adenocarcinomas compared with either the same patients after 6 weeks of CPAP or matched control subjects without OSA [140]. The mechanisms by which exosomes accelerate the growth and invasiveness of malignant tumors are partially understood [140]. It has been shown that exosomes inhibit the number and cytotoxic activity of NK cells [140]. Interestingly, this effect appears to be mediated by the PGE_2_/EP2/EP4 signaling pathway [141]. Blocking PGE_2_ synthesis by indomethacin, and EP2 or EP4 by specific receptor antagonists, attenuated the inhibitory capacity of exosomes [141]. Tumor exosomes also promote the induction of MDSC [142]. Interestingly, MDSC-mediated promotion of tumor progression and invasiveness is also dependent on the PGE_2_ present in tumor exosomes [142].

The COX-2/PGE_2_ pathway is also vital for activation of canonical Wnt/β-Catenin signaling and its inhibition by COX-2 inhibitors attenuates the signaling transmission and restricts tumor growth [143]

Collectively, these observations add further support to the important role played by the COX-2/PGE_2_ axis in the regulation of the multiple mechanisms involved in tumorigenesis and suggest the necessity of exploring its role as link between OSA and cancer (Appendix A).

This was the objective of a very recent study in which the contribution of the COX-2/PGE_2_ pathway in IH-induced enhanced tumor malignancy was assessed using celecoxib as a COX-2 specific inhibitor in a murine model of OSA bearing LLC1 tumors [144]. IH exposure promotes accelerated tumor growth when compared to control normoxic mice. Daily administration of celecoxib (75 mg/Kg body weight) significantly decreased IH-induced tumor growth (mean weight in untreated 2.07 g ± 0.28 vs. mean weight in treated 0.95 g ± 0.15). In absence of celecoxib treatment, a significant increase in TAMs, MDSCs, and Tregs was found in tumors from mice exposed to IH compared with the control group. In contrast, in IH-exposed mice treated with celecoxib, TAMs, MDSCs, and Tregs counts were restored to values found in control normoxic mice. In addition, IH exposure induced a polarity shift in TAMs from an anti-tumoral phenotype (M1) towards a tumor-promoting phenotype (M2). These changes were associated with down-regulation in the expression of the M1 marker INF-γ, while the expression of IL-10, an M2 marker, was up-regulated. All these changes were reverted by celecoxib treatment to levels similar to those found in control normoxic mice. As expected, celecoxib treatment significantly inhibited PGE_2_ production by tumor cells and TAMs [144].

The results of this study provide support to the hypothesis that the up-regulation of the COX-2/PGE_2_ pathway induced by hypoxia plays a central role in the association of OSA and cancer.

Despite the proven effect of aspirin to prevent the development and progression of cancer, none of the studies published so far to investigate the association of OSA and cancer have taken into account the use of aspirin in the analysis. Since a high percentage of patients with OSA suffer from cardiovascular and neurovascular diseases, it is expected that a large number of patients included in the published studies would be receiving low doses of aspirin. Does the use of low dose of aspirin protect OSA patients from developing cancer? The answer to this question remains to be investigated.

## 7. Conclusions

Studies carried in animal models have shown that IH induces changes in several signaling pathway and transcription factors involved in the regulation of host immunological surveillance that finally results in tumor establishment and invasion. It has also been shown that IH induces the expression of COX-2 that results in an increased synthesis of PGE_2_.

Relevant scientific evidence supports the central role played by the COX-2/PGE_2_ axis in oncogenesis. The fact that this axis is induced by IH in patients with OSA, supports the hypothesis that OSA and cancer are pathogenically related, at least in part, by the activation of the COX-2/PGE_2_ pathway. A recent study conducted in a mouse model demonstrated that the blockage of the COX-2-PGE_2_ pathway through a specific COX-2 inhibitor (celocoxib) was able to abrogate the pro-oncogenic effects of IH. These findings are in line with epidemiological studies showing the preventive effect of aspirin (both COX-1 and COX-2.

Several studies evaluating the association between OSA and cancer have reported discrepant conclusions. Some studies have shown that OSA increases the incidence of malignant tumors; however, there are also studies that have failed to find such an association. Other studies have found that the incidence of some cancers increases in patients with OSA (e.g., Pancreas and melanoma), while the incidence of other malignant tumors appears to decrease in the same patients (e.g., Colorectal and breast).

Although the reasons for these differences remain to be elucidated, variation in the way IH is assessed, length of observation periods, and the contribution of confounding factors not included in the study, can probably account for the discrepancies

The contradictory results reported in epidemiological studies on the association between OSA and cancer reveal the complexity of the problem. Despite the ability of aspirin to prevent the development and progression of cancer, none of the studies to investigate the association of OSA and cancer have included the use of aspirin as a potential preventive therapy in the analysis.

A high percentage of patients with OSA suffer from cardiovascular and neurovascular diseases that are usually treated with low dose aspirin for preventive purposes. The use of aspirin by these patients can mitigate the pro-oncogenic effects of IH-induced excessive PGE_2_ synthesis in OSA patients. Further studies are needed to investigate the regulation of the COX-2/PGE_2_ pathway and the impact of aspirin in the prevention of malignant tumors patients with OSA.

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
