# Peer review of "Role of the Cyclooxygenase Pathway in the Association of Obstructive Sleep Apnea and Cancer"

_jcm, 2020, doi:10.3390/jcm9103237_

Round 1
Reviewer 1 Report
The authors reviewed about the relevance between OSA and cancer with the aspects of epidemiological studies and experimental studies used animal model. Although the manuscript consisted of 5 sections and several sub sections, some of the section titles were inappropriate for the contents. For instance, the tile of section 3; “COX pathway and cancer” mentioned only about COX pathway but didn’t cancer. Moreover, there were several typo.
Therefore, the authors should read the manuscript from the head and reconsidering the section titles and contents. And a number of points need clarifying and certain statements require further justification. These are given below.
- Although the manuscript consisted of 5 sections and several sub sections, some of the section titles were inappropriate for the contents. For instance, the tile of section 3; “COX pathway and cancer” mentioned only about COX pathway but didn’t cancer.
- The reference of page 5, line 2 was incorrect. The paper was no mentioned about the Tregs responses.
- There are so many sentences where review articles are cited. Please do not cite the review article, try to cite the original paper.
- Several brackets for referencing were written with the parentheses. Did you use reference management software? Please check all references, and I strongly recommend using some software for reference management.
Typographical errors listed below should be corrected:
Page 2: (95% CI, 1.3-2.9; p=0.002) -> (95% CI, 1.3-2.9; p=0.002)
* As the letter of “p” stands for the initials of the probability, the letter should be written as italic.
Page 3: (n=700.000) -> (n=700,000)
Page 3: (n=30.000) -> (n=30,000)
Page 5: the regulatory T cells (Tregs) -> the Tregs
* The abbreviation was already written at page 4.
Page 7: M 1(Th1 response) or to M 2 (Th 2 response) -> M1(Th1 response) or to M2 (Th2 response)
Page 7: natural killer T cells -> NK cells
* The abbreviation was already written at page 4.
Page 8: programmed death-ligand 1 (PD-L1) -> PD-L1
* The abbreviation was already written at page 6.
Page 9: (COX-1)-and -> (COX-1) and
Page 15: CPAP
* I can’t find the abbreviation.
Page 16: IH exposure
* I can’t find the abbreviation.
Author Response
Please seethe attachment

Reviewer 2 Report
In this interesting manuscript, Picado and Roca-Ferrer review the potential relationship between obstructive sleep apnea (OSA) and cancer and highlight the overexpression of Cyclooxygenase as a possible common thread. I have only a few comments to improve upon the quality of this work.
- There is undoubtedly a lack of figures. I think a summary figure covering all the points discussed in the immune response, maybe with pro and anti-tumoral effect, highlighting the action in a normal and a tumor environment would improve clarity. Similarly, a figure that summarizes the potential action of the prostaglandins (PG) on a tumor cell in OSA and non-OSA patients (possibly with the sites of action of aspirin) would be welcome.
- I think the paper of Azarbarzin et al on the European Heart Journal, which did not describe any association between hypoxic burden and other-than-cardiovascular causes of mortality (including cancer) in OSA is worth a mention in the introduction. In the same paragraph the authors should comment on the fact that many studies did not find a real link between OSA and cancer and that the risk of some cancers diminished in OSA patients.
- The authors should clarify better the “immune equilibrium” step of the carcinogenesis progression as, for how they describe it, it seems too similar to the escape phase.
- The authors state that “the progression of human cancer is associated with a shift in Th1 to Th2 immune,” but did not look into why a Th1-to-Th2 shift would be deleterious to contrast tumor response.
- When the Tregs are mentioned for the first time, I think that detailing their function (similarly to what the authors had done for the other components of the immune response) would help the reader. Likewise, the authors should consider adding the way CD8+ induces elimination of tumor cells. In this paragraph, “point mutations in normal genes” needs further explanation; genes coding for what?
- When talking about the dendritic cells, there is a continuous back and forth of pros and cons (and this happens other times throughout the manuscript). I encourage the authors to proceed with a more ordinated list of beneficial and deleterious factors/effects.
- In the macrophages paragraph, the authors should consider adding a few lines explaining how the macrophages get recruited in the tumor mass and how the switch from M1 to M2 happens. In this same paragraph, the authors use TAMS as acronym, typo for TAM? Other acronyms should be decoded upfront (e.g. HIFs, PKA)
- In the MDSC paragraph, I think “cells” is missing after “these immune-suppressive in tumors”. Also note that there is no need to repeat the definition/role of MDSC (or any other previously defined word) when describing it more in details ahead in the text (e.g. no need to repeat MDSC function in the exosome part; the Wnt/β-Catenin signaling pathway is also mentioned prior and after the exosome part).
- The part describing the hypoxia inducible factors is a poorly detailed to me and there are many studies in literature encompassing the links between OSA and HIFs. Since this is a key message of the manuscript, I suggest adding more information here.
- The authors should justify why they talk in detail only about PG E and D.
- The authors state that “PGE inhibits proliferation and effectors functions of CD4+ and CD8+ T cells and promotes their differentiation in Treg”, however, to my knowledge, it improves Th2 function.
- The authors state that PGE “promotes tumor growth by suppressing tumor-infiltrating DCs activation”. Again, to my knowledge, PGE2 supports activation of dendritic cells, but suppresses their ability to attract naive, memory and effector T cell.
- While reading the manuscript, I realized that, many times, it was not clear if the mentioned studies have been performed in animal or human models. Please specify when needed.
- Why do the authors think that only aspirin, and not another NSAID, would be a good candidate to prevent cancer in OSA patients?
- After reference 145, “promoted” is a typo.
- References 132-138 should go before 136,137.
- A minor English spelling check is needed.
Reviewer 3 Report
The authors presented a review regarding the potential association between apnea and cancer, with a clear focus on the immunological aspect of cancer.
I don't have any significant comments. There were a few mistakes that need to be edited:
1 -Abstract needs to have at least a sentence about the purpose of this review.
2- There are some of the reference numbers in different fonts.
3- The conclusion needs to be revised. It is a copy-paste from the abstract and the text.
4-Please provide a reference for this statement “In addition, recent observations support that the excess in mortality detected in OSA may also be the consequence of an increased incidence of malignant tumors in these patients.”
5- Page 9, Angiogenesis refers to the physiological process in which new blood cells grow from pre-existing vessels [67]. It should be new blood vessels, not blood cells. Please correct this statement.
6- Since the review has two authors, please separate the name by "and" instead of ","
Author Response
Please seethe attachment

Round 2
Reviewer 1 Report
The words and sentences were seem to have been polished and I felt great passion of the authors for this review article.
The manuscript can be accept in the present form I think.